# Right Ventricular Strain by Magnetic Resonance Feature Tracking Is Largely Afterload-Dependent and Does Not Reflect Contractility: Validation by Combined Volumetry and Invasive Pressure Tracings

**DOI:** 10.3390/diagnostics12123183

**Published:** 2022-12-16

**Authors:** Andreas Rolf, Till Keller, Jan Sebastian Wolter, Steffen Kriechbaum, Maren Weferling, Stefan Guth, Christoph Wiedenroth, Eckhard Mayer, Christian W. Hamm, Ulrich Fischer-Rasokat, Julia Treiber

**Affiliations:** 1Kerckhoff Heart and Thorax Center, Department of Cardiology, Benekestr. 2-8, 61231 Bad Nauheim, Germany; 2Medical Clinic I, Department of Cardiology, University of Giessen, 35390 Giessen, Germany; 3German Center for Cardiovascular Research (DZHK), Partner Site RheinMain, 61231 Bad Nauheim, Germany; 4Kerckhoff Heart and Thorax Center, Department of Thoracic Surgery, 61231 Bad Nauheim, Germany

**Keywords:** pulmonary hypertension, right ventricle, remodeling, afterload, contractility, strain

## Abstract

Cardiac magnetic resonance (CMR) is currently the gold standard for evaluating right ventricular (RV) function, which is critical in patients with pulmonary hypertension. CMR feature-tracking (FT) strain analysis has emerged as a technique to detect subtle changes. However, the dependence of RV strain on load is still a matter of debate. The aim of this study was to measure the afterload dependence of RV strain and to correlate it with surrogate markers of contractility in a cohort of patients with chronic thromboembolic pulmonary hypertension (CTEPH) under two different loading conditions before and after pulmonary endarterectomy (PEA). Between 2009 and 2022, 496 patients with 601 CMR examinations were retrospectively identified from our CTEPH cohort, and the results of 194 examinations with right heart catheterization within 24 h were available. The CMR FT strain (longitudinal (GLS) and circumferential (GCS)) was computed on steady-state free precession (SSFP) cine CMR sequences. The effective pulmonary arterial elastance (Ea) and RV chamber elastance (Ees) were approximated by dividing mean pulmonary arterial pressure by the indexed stroke volume or end-systolic volume, respectively. GLS and GCS correlated significantly with Ea and Ees/Ea in the overall cohort and individually before and after PEA. There was no general correlation with Ees; however, under high afterload, before PEA, Ees correlated significantly. The results show that RV GLS and GCS are highly afterload-dependent and reflect ventriculoarterial coupling. Ees was significantly correlated with strain only under high loading conditions, which probably reflects contractile adaptation to pulsatile load rather than contractility in general.

## 1. Introduction

Pulmonary hypertension (PH) is a progressive and life-threatening disorder that potentially complicates many cardiovascular and respiratory diseases [1]. Increasing pulmonary pressure overloads the right ventricle (RV), causing adverse remodeling of the RV and leading to right heart failure and ultimately death [2,3]. Although the increase in pulmonary artery pressure is the distinctive characteristic of the disease, it is the RV function that governs long-term prognosis.

Cardiac magnetic resonance imaging (CMR) has emerged as the gold-standard method for evaluating RV function, as it allows full coverage of the complex RV geometry with superior image quality and provides low intra- and interobserver variability with good interstudy reproducibility [4,5,6,7,8]. In addition to allowing volumetric measurements of the RV ejection fraction (EF), CMR feature-tracking (FT) strain analysis is a promising new technique that can be used to quantify myocardial dysfunction [4,9].

Strain analysis has several potential advantages over traditional indices of RV function. It can detect subtle changes in ventricular function before EF is reduced [10,11,12]. Furthermore, strain measures seem to have independent prognostic value and are able to predict mortality better than established parameters such as RV-EF or tricuspid annular plain systolic excursion (TAPSE) [13,14,15,16,17,18,19,20,21]. However, the physiological correlation of RV strain still remains to be elucidated.

There is conflicting evidence as to the load dependence of strain and its relationship to contractility [22]. While there is growing agreement on the afterload dependence of strain, contractility seems to influence strain but is not independent of loading conditions, as Ferferieva et al. and Weidemann et al. have shown for the left ventricle using echo-derived strain [23,24]. In a recent publication, Tello et al. used FT strain to analyze both RV function as well as afterload in relation to contractility using CMR and conductance catheter indices. They found that strain is strongly afterload-dependent in PH patients and is not an index of contractility [25].

In previous work, we showed that the conductance catheter indices of afterload, contractility, and ventriculoarterial coupling (Ea, Ees, and Ees/Ea) could reliably be approximated using right heart catheter (RHC)-derived mean pulmonary artery pressure (mPAP) and volumetry data obtained by CMR to monitor RV afterload and function [26].

It was the purpose of this study to extend the findings of Tello et al. by correlating global longitudinal and circumferential RV strain (GLS and GCS) with Ea, Ees, and Ees/Ea under two different afterload conditions before and after pulmonary endarterectomy (PEA) by using CMR and RHC data from patients with chronic thromboembolic pulmonary hypertension (CTEPH). In contrast to idiopathic pulmonary arterial hypertension (PAH), CTEPH represents an “on-off” model of PH as it is potentially curable by PEA or balloon angioplasty.

## 2. Materials and Methods

### 2.1. Patients and Ethics Statement

From our in-house Bio-CVI imaging registry, we retrospectively identified 496 patients who underwent CMR before PEA, immediately after PEA, or 12 months after PEA between the years 2009 and 2022. Of the 601 CMR examinations that were available from that period, 268 had no matching RHC; of the remaining 333 examinations, 142 patients with 194 examinations were identified for whom RHC data were available within 24 h of the CMR examination. All CMR examinations were routinely performed for assessment of RV function at the discretion of the thoracic surgeon in the work-up or follow-up of CTEPH patients. Contraindications for CMR were renal failure with glomerular filtration rate below 30 mL/min/1.73 m^2^, incompatible metallic implants, known intolerance to gadolinium, and claustrophobia. 

All patients gave written informed consent. The study was approved by the ethics committee of the University of Giessen (AZ 199/15). Compare Figure 1. 

### 2.2. Hemodynamic Background and Formulas

Elastance is generally defined as the change in pressure for a given change in volume [27]. The higher the pressure the RV can generate at a given end-systolic volume (RVESVi), the better its contractility. The end-systolic pressure/volume ratio Ees of the RV is therefore a measure of contractility. It can be reliably approximated by mPAP (measured by routine RHC) divided by RVESVi, which is measured by volumetry of the RV and indexed to body surface area [28]:Ees = mPAP/RVESVi(1)

Conversely, the higher the end-systolic pressure of the pulmonary artery (PA) for a given stroke volume indexed to body surface area (RVSVi), the higher the resistance and impedance and the lower the compliance of the pulmonary vasculature. The end-systolic pressure/volume ratio Ea of the PA is therefore a surrogate marker for afterload. It can be approximated as mPAP divided by RVSVi [28]:Ea = mPAP/RVSVi(2)

Optimal coupling of the PA is achieved if the RV is able to generate maximal flow with minimal energy loss, which is possible when there is maximal transfer of energy from one elastic chamber (the ventricle) to another (the arterial system), as is the case when the ventricular-arterial coupling ratio (Ea/Ees) is between 1 and 2 [29].

### 2.3. CMR Acquisition

All patients were examined on a 1.5 T MR scanner (Sonata, Siemens, Erlangen, Germany) or a 3T MR scanner (Skyra, Siemens, Erlangen, Germany) in the head-first, supine position using a six-channel/18-channel phased array surface coil. Typical steady-state free precession (SSFP) cine CMR sequence parameters were TE 1.5 ms, TR 38.8 ms, 13 segments, 1.6 × 2.2 mm in-plane resolution, flip angle 70°, bandwidth 930 Hz/px, slice thickness 8 mm, and interslice gap 2 mm.

Volumetric measurements of RV function were performed in a standard fashion by cine CMR covering the RV in short-axis slices from base to apex. Endocardial and epicardial contours were drawn on the RV to determine end-diastolic, end-systolic, and stroke volumes, as well as RV myocardial mass (RVMASS) and EF, using cvi42 software (circle cardiovascular imaging, Calgary, AB, Canada). Trabeculations were excluded from the myocardium. All volumetric parameters were normalized for body surface area (EDVi, ESVi, RVMASSi, SVi).

### 2.4. Feature Tracking

RV and systolic strain rates were calculated using the FT module from cvi42 (circle cardiovascular imaging, Calgary, AB, Canada). First, the RV endocardial and epicardial borders of the RV free wall at the end-diastolic (ED) phase in short-axis and 4-chamber long-axis cine images were traced. The interventricular septum was not included, and RV trabeculations were carefully excluded. The software automatically propagated contours throughout all phases. The propagated myocardial tissue across the cardiac cycle was verified by the operators to ensure the accuracy of the propagation. Global strain (in the longitudinal and circumferential direction) in the short-axis and long-axis views was then automatically derived by the software. RV global circumferential strain (GCS) was obtained using short-axis cine views. RV global longitudinal strain (GLS) was obtained from a 4-chamber long-axis view. See Figure 2 for GCS images.

All measurements (CMR and FT) were performed independently by two experienced cardiologists (J.V. and A.R. with 5 and 16 years of experience, respectively).

### 2.5. Right Heart Catheterization

RHC measurements were performed using standard Swan-Ganz catheters introduced via a 6 F sheath through the internal jugular, subclavian, or femoral vein. Measurements were obtained from RHC procedures during a pre- or postoperative routine check-up. RHC and CMR were performed within 24 h of each other. Patients were not treated with vasoactive agents during RHC measurements.

### 2.6. Statistics

The Shapiro–Wilk test was performed to test for the normal distribution of the data. Continuous data following a normal distribution are presented as mean ± SD, and continuous data not normally distributed are presented as median and IQR. Categorical variables are given as absolute counts and frequencies. Spearman´s rank correlation coefficient (rho) was used to test the correlation between hemodynamic measurements (Ea, Ees, Ea/Ees), mPAP, PVR, and GLS/GCS. Differences between patients examined before and after PEA were tested with Student’s *t*-test or Wilcoxon’s rank sum test, where appropriate. Scatter lots with line fit were used to visually display the relationship between strain and hemodynamics. To test for an interaction between GLS or GCS and whether measurements were made before or after PEA, linear regression was calculated after the log transformation of Ea and Ees. A *p*-value of less than 0.05 was considered statistically significant. All tests were computed using STATA17 (StataCorp, College Station, TX, USA).

## 3. Results

### 3.1. Patient Characteristics

A total of 142 patients with 194 examinations were retrospectively enrolled after a search of the registry; 72 of the examinations were prior to PEA, and 122 were after PEA, of which 19 were during the postoperative hospital stay and 103 at the 12-month follow up. The mean age was 57.8 (45–71.3) years, and 93 (48%) were female. The baseline characteristics, CMR measurements, and hemodynamic data are displayed in Table 1.

The strain was significantly improved in patients examined after PEA than in those examined before PEA. Afterload (Ea) and contractility (Ees) were lower after PEA, showing that ventricular adaptation to increased afterload and coupling was reversible, as expected. PVR and mPAP were significantly lower in patients after PEA; full details are provided in Table 2.

### 3.2. Association between Strain and Physiological Parameters of Afterload, Contractility, and Coupling

GLS was significantly correlated with Ea and Ea/Ees but not with Ees. The strongest correlation was found for ventriculoarterial coupling as represented by Ees/Ea (rho −0.56, *p* = 0.00001). A slightly weaker correlation was only found for Ea (rho 0.5, *p* = 0.00001). GCS was also significantly correlated with Ea and Ees/Ea but not with Ees; the coefficients were similar (rho 0.49, *p* = 0.0001; rho −0.61, *p* = 0.00001, respectively; for details, see Table 3 and Figure 3). For patients before PEA, the association of Ees with both longitudinal and circumferential strains was significant, and the p for interaction was also significant; this was especially relevant for GLS (Figure 4). Generally, the associations were stronger in examinations before PEA than after PEA for both types of strain.

### 3.3. Association between Strain and Hemodynamic/Volumetric Data or NT-proBNP

GLS was significantly correlated with mPAP and PVR (rho 0.47, *p* = 0.0001; rho 0.52, *p* = 0.0001), as was GCS (rho 0.48, *p* = 0.0001; rho 0.48, *p* = 0.0001, respectively; Figure 3). GLS was likewise significantly correlated with RV-EDVi and NT-proBNP (rho 0.23, *p* = 0.003; rho 0.45, *p* = 0.0001, respectively). Similar correlations were found between GCS and RV-EDVi or NT-proBNP (rho 0.27, *p* = 0.0005; rho 0.37, *p* = 0.0001, respectively; Figure 5).

## 4. Discussion

The present study examined the relationship between strain and its physiological determinants—afterload and contractility—in a cohort of CTEPH patients. Patients were examined before and after PEA, which allowed an assessment of the relationship between strain, arterial load, and contractility, as well as a ventriculoarterial coupling in two different (high and low) afterload conditions. Afterload and contractility were approximated as the effective arterial and chamber elastance, Ea and Ees, respectively, by combining invasive pressure tracings in the PA and volume data by CMR, as originally proposed by Kuehne et al. and confirmed by Sanz et al. [27,28].

The main findings of our study are:(1)RV longitudinal and circumferential strain correlated well with afterload as represented by Ea in both afterload conditions (before and after PEA);(2)RV longitudinal and circumferential strain did not correlate with contractility in the whole cohort; however, there was a significant and modest correlation in the setting of elevated pulmonary pressure before PEA. After PEA, there was no correlation between contractility and strain in CTEPH patients;(3)Longitudinal and circumferential strain correlated well with ventriculoarterial coupling both before and after PEA.

Thus, the main determinant of right ventricular strain is pulmonary afterload. A moderate relationship with contractility only becomes evident if contractility is maximally adapted to increased afterload.

End-systolic pressure relationships measured by conductance catheters were first introduced for left ventricular measurements by Sagawa et al. and later applied to the right ventricle [30,31]. The effective elastance of the PA, Ea, reflects afterload better than PVR as it also incorporates pulsatile load (impedance), which is especially important in CTEPH patients with early wave reflections of the vasculature [32]. Kuehne et al. were the first to show that it is possible to approximate conductance measurements closely by simply using RV volumes from CMR examinations and mPAP obtained by RHC [28]. In previous work, we have employed this method to show that afterload conditions significantly improve in CTEPH patients after PEA [26].

CMR has become the gold standard in evaluating RV function, and the additional use of FT allows the quantification of longitudinal and circumferential strain for an in-depth study of RV fiber motion [6,33,34]. Tello et al. recently published data on the correlation of strain with conductance catheter-derived Ees, Ea, and Ees/Ea and found that strain predicts afterload (Ea) and ventriculoarterial coupling (Ees/Ea) but not contractility (Ees) [25]. The correlation coefficients determined in our study are in very good agreement with those found by Tello et al., which supports the validity of our approximation method. Our data also confirm the strong afterload dependence of RV strain and its relation to coupling under two different afterload conditions, before and after PEA. The relationship between strain and Ea (as an afterload parameter) and strain and Ees/Ea (as a marker of RV/PA coupling) is maintained after pulmonary circulation is restored. Hence, the relationship between strain and afterload and coupling is significant in loaded and unloaded ventricles, which extends and generalizes the findings of Tello et al.

However, there was also a significant association of GCS and GLS with Ees that was not described by Tello et al. [25]. Of note, this relationship is only evident if the analysis is confined to patients before PEA under conditions of high afterload, and it is diminished if patients with restored pulmonary vasculature post-PEA are included. Tello et al. hypothesized that Ees in their study was already maximally increased in compensation for elevated afterload and cannot adapt further; hence, the lack of correlation between Ees and strain [25]. This hypothesis is supported by data from Sanz et al., who showed that Ees increases in response to increased afterload and decreases again with further rising PVR. In their study, Ees increased in the first to third quartiles and decreased in the fourth above 8 WU [27].

In previous work, we showed that Ees is increased before PEA in compensation for increased afterload and decreases afterwards, which is consistent with the results obtained in this larger cohort [26]. Considering the whole cohort of the present study, the Ees and PVR relationships are similar to those described by Sanz, with increasing Ees up to the fourth quartile, which begins at 6.78 WU [27] (Appendix A). In the group of patients in the high-afterload setting before PEA, there is no dose–response relationship between PVR quartiles and Ees (Appendix A). This supports Tello’s theory of a maximally adapted Ees that cannot increase further. However, it is surprising that a significant relationship between strain and Ees was observed in our cohort despite this maximally adapted contractility.

We can only speculate about an explanation for this phenomenon. The large majority of patients in Tello’s study were PAH patients; only 16% had CTEPH. The mechanism of increased pulmonary resistance is different between these groups, with elevated upstream resistance in CTEPH patients and downstream resistance in PAH patients. The large increase in upstream resistance causes early wave reflections in the vascular bed that over-proportionately increase the pulsatile load of the ventricle. This pulsatile component is not well represented by PVR alone. The mean PVR in the group of patients before PEA was 7.7 (6.6) dyn/s/cm^−5^, which was markedly higher than in the Tello cohort, but the mean Ea was even more elevated compared with that of Tello’s patients (0.97 instead of 0.7 mmHg/mL) [25]. This reflects the importance of pulsatile load in CTEPH. Accordingly, a dose–response relationship can be observed for Ees across quartiles of Ea in the pre-PEA group (Appendix A); thus, while Ees does not further adapt to PVR, it does change in response to increasing pulsatile load as represented by Ea. Even though this is only speculative, it is supported by data from Guihaire et al., who also found a significant relationship between Ees and conventional echocardiographic parameters of contractility such as the RV myocardial performance index (RVMPI) and acceleration during isovolumetric contraction (IVA) in a porcine model of pulmonary hypertension [35]. Interestingly, they induced PH by ligature of the PA in succession with repeated embolism, which represents a mechanism similar to that occurring in CTEPH.

### Limitations

The findings of this study are based on surrogate markers of Ea and Ees and not on the gold standard of conduction catheter measurements. However, these markers have been thoroughly validated previously. To ensure a good approximation of Ea and Ees, only patients with available RHC and CMR data within 24 h were included. Additionally, since, to the best of our knowledge, this is the largest cohort of CTEPH patients in which hemodynamic and strain data are correlated in the setting before and after PEA under different loading conditions, it appears to be justified to make inferences regarding the load dependence of the RV strain in general.

## 5. Conclusions and Clinical Perspective

Our findings demonstrate that RV strain is largely governed by afterload and ventriculoarterial coupling, which is in line with results from previous studies. However, in CTEPH patients with high pulsatile load before PEA, the RV strain also seems to reflect the adaptation of Ees to increasing pulsatile loading conditions. This could be shown without the use of costly and not generally available PV-loop measurements.

These results also highlight that strain reflects hemodynamic changes that could otherwise only be measured invasively, with increased risk for the patient. Thus strain, more than RV-EF alone, should be used more widely to measure RV function and especially monitor changes in RV function over time.

For this paper, we used CMR and RHC measurements before and after PEA in a large cohort of patients. However, only 18 patients could provide measurements at all three time points before, immediately after and 12 months after PEA. In the future, we aim to reproduce these data in a cohort of patients with data for all visits before and after PEA to have a more in-depth understanding of how strain develops over time and after intervention.

## Figures and Tables

**Figure 1 diagnostics-12-03183-f001:**
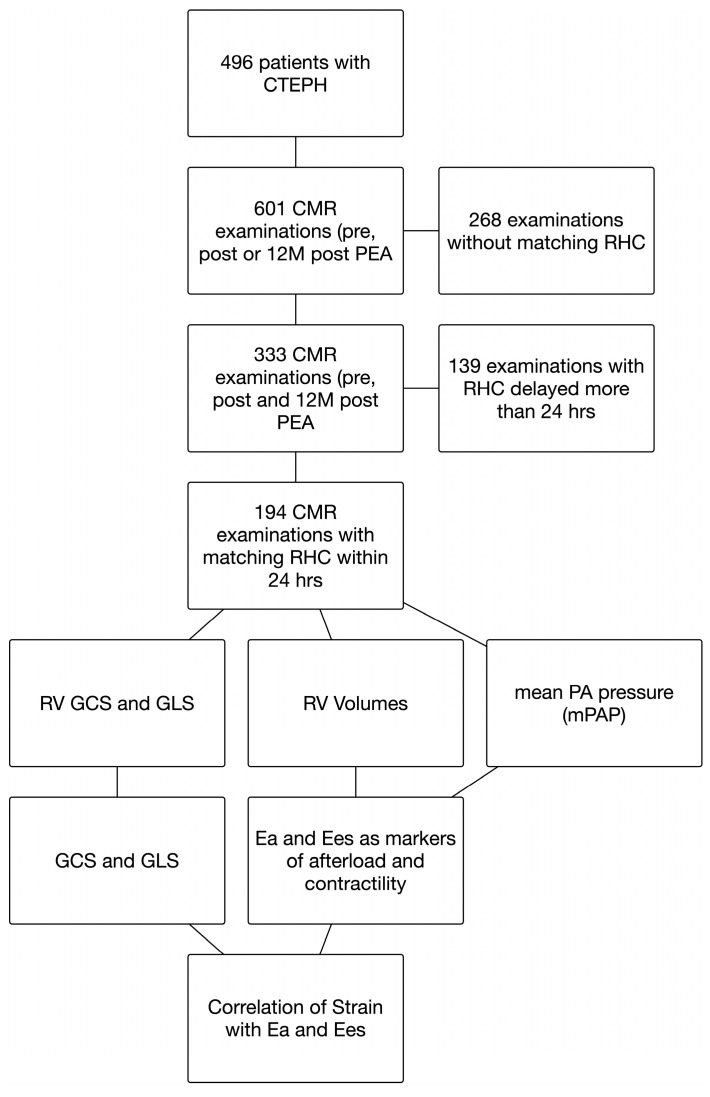
Patient selection and analysis procedure. CMR—cardiac magnetic resonance, RHC—right heart catheter, GCS—global circumferential strain, GLS—global longitudinal strain, RV—right ventricle, Ea—effective arterial elastance, Ees—effective elastance of the RV, PA—pulmonary artery.

**Figure 2 diagnostics-12-03183-f002:**
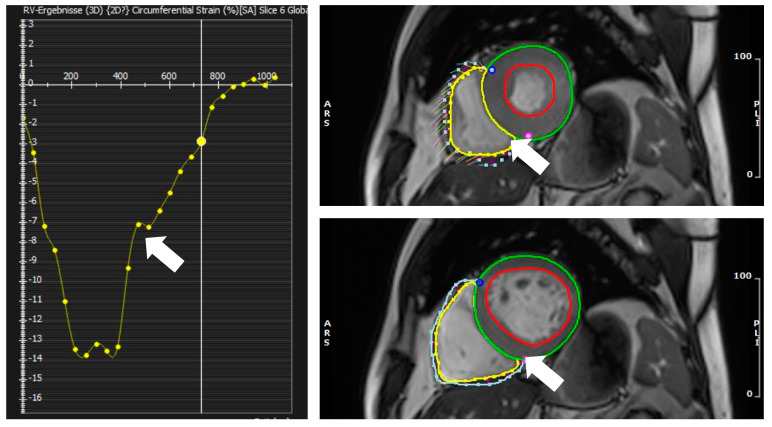
(**Left panel**) shows global circumferential strain GCS with the strain curve over the cardiac cycle (white arrow). (**Right upper panel**)—needles show the movement of features in systole (white arrow); (**Right lower panel**) user defined diastolic contours of the RV from which features are extracted (white arrow).

**Figure 3 diagnostics-12-03183-f003:**
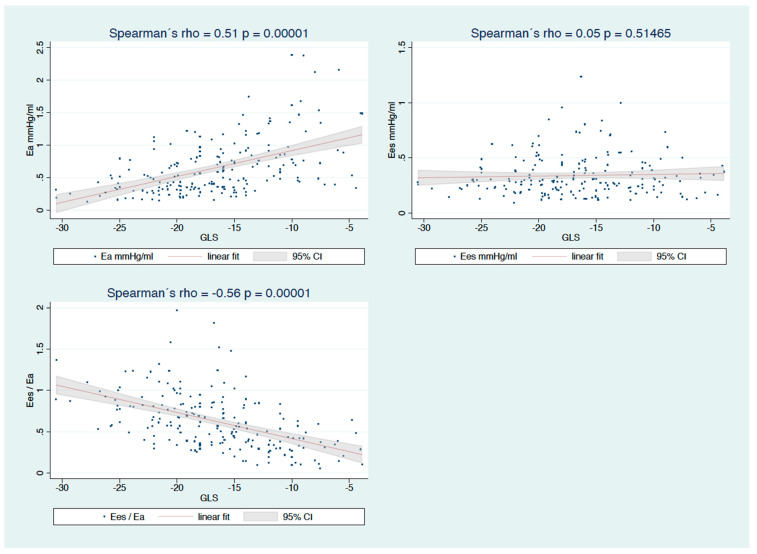
Correlations of GLS with afterload Ea (pulmonary arterial elastance), contractility Ees (end-systolic elastance of the RV), and ventriculoarterial coupling Ees/Ea.

**Figure 4 diagnostics-12-03183-f004:**
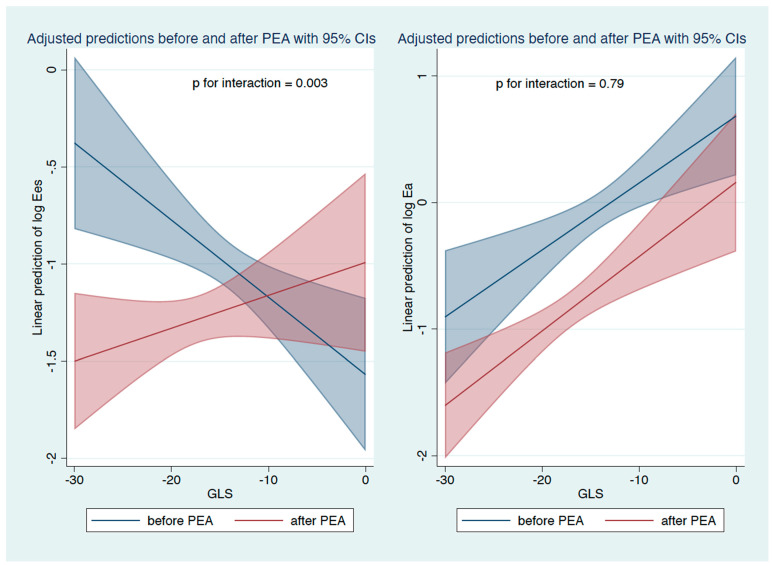
Linear relationship between GLS and contractility Ees (end-systolic elastance of the RV) and its interaction with PEA—(**left panel**). Linear relationship between GLS and afterload Ea (elastance of the pulmonary artery) and its interaction with PEA—(**right panel**).

**Figure 5 diagnostics-12-03183-f005:**
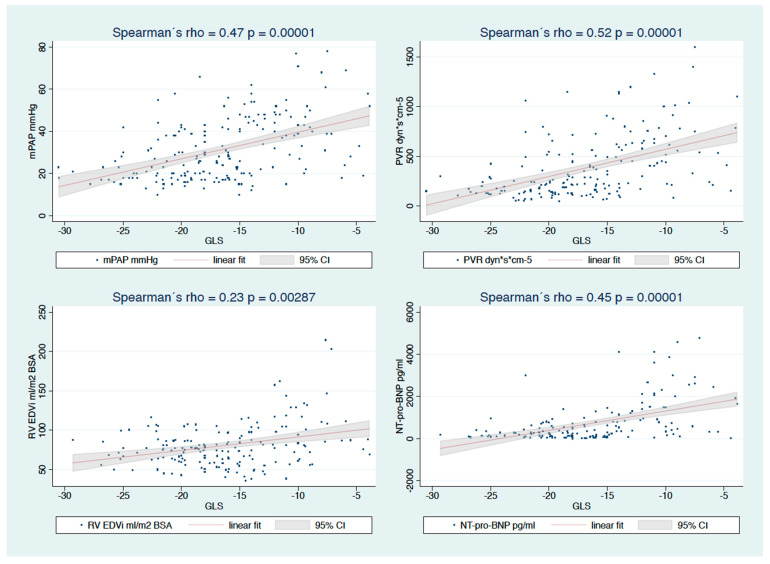
Correlations of GLS with mPAP, mean arterial pressure of the pulmonary artery, PVR, pulmonary vascular resistance, RV EDVi, right ventricular end-diastolic volume indexed for body surface area and NT-pro BNP.

**Table 1 diagnostics-12-03183-t001:** Patient characteristics.

Patient Characteristics	Value ^1^
General	
Age, years	57.8 (45–71.2)
Female sex	93 (48%)
NYHA class ≥ III ^2^	65 (38%)
6 min walking distance, m	438 (306–530)
NT-proBNP, pg/mL	323 (117–864)
Hemodynamics	
mPAP, mmHg	27 (19–42)
PVR, dyn/s/cm^−5^	252.5 (150–542)
Physiological data	
Ea, mmHg/mL	0.52 (0.33–0.91)
Ees, mmHg/mL	0.29 (0.22–0.43)
Ees/Ea ratio	0.57 (0.37–0.84)
CMR strain	
GLS, %	−16.7 ± 5.5
GCS, %	−12.1 ± 3.9
CMR volumetry	
RV EDVi, mL/m^2^	77.1 (62–96.9)
RV ESVi, mL/m^2^	44.6 (35.8–60.1)
RV SVi, mL/m^2^	27.4 (21.6–34.8)
RV EF, %	37 (27–46)
RVMASS, g	53 (21–81.3)

^1^ Values given as number (%), mean ± SD, or median (IQR); ^2^ Abbreviations: CMR, cardiac magnetic resonance; Ea, pulmonary arterial end-systolic pressure/volume ratio; EDVi, end-diastolic volume indexed to body surface area; Ees, right ventricular end-systolic pressure/volume ratio; EF, ejection fraction; ESVi, end-systolic volume indexed to body surface area; GCS, global circumferential strain, GLS, global longitudinal strain; mPAP, mean pulmonary artery pressure; NT-proBNP, N-terminal fragment of pro-brain natriuretic peptide; NYHA, New York Heart Association; PVR, pulmonary vascular resistance; RV, right ventricle; RVMASS, right ventricular mass; SVi, stroke volume indexed to body surface area.

**Table 2 diagnostics-12-03183-t002:** Differences between patients scanned before and after PEA.

Patient Characteristics	Before PEA ^1^	After PEA	*p*-Value
General			
Age, years	57.9 (45.5–71.5)	57.6 (44.6–70.9)	0.83
Female sex	78 (44%)	56 (46%)	0.54
NYHA class ≥ III ^2^	63 (88%)	2 (2%)	0.0001
6 min walking distance, m	390 (285–470)	480 (385–546)	0.0003
NT-proBNP, pg/mL	801 (280–1583)	226 (101–489)	0.00001
Hemodynamics			
mPAP, mmHg	43 (38–50)	20 (17–27)	0.00001
PVR, WU	7.7(5.6–9.8)	2.2 (1.6–3.1)	0.00001
Physiological data			
Ea, mmHg/mL	0.97 (0.68–1.43)	0.37 (0.28–0.54)	0.00001
Ees, mmHg/mL	0.37 (0.26–0.5)	0.27 (0.19–0.36)	0.0001
Ees/Ea ratio	0.37 (0.26–0.6)	0.68 (0.5–0.88)	0.00001
CMR strain			
GLS, %	−13.9 ± 4.6	−18.3 ± 5.4	0.00001
GCS, %	−9.2 ± 3.4	−13.7 ± 3.1	0.00001
CMR volumetry			
RV EDVi, mL/m^2^	82.2 (62–105)	75.6 (61–86.2)	0.049
RV ESVi, mL/m^2^	57.3 (38.7–79.1)	41.8 (35–50.3)	0.0001
RV SVi, mL/m^2^	25.6 (20.3–30.3)	28.6 (22.1–37.3)	0.02
RV EF, %	27 (15.5–37)	41 (35–48)	0.00001
RVMASS, g	81.5 (66–99.5)	23 (18–41.8)	0.00001

^1^ Values given as number (%), mean ± SD, or median (IQR); ^2^ Abbreviations: CMR, cardiac magnetic resonance; Ea, pulmonary arterial end-systolic pressure/volume ratio; EDVi, end-diastolic volume indexed to body surface area; Ees, right ventricular end-systolic pressure/volume ratio; EF, ejection fraction; ESVi, end-systolic volume indexed to body surface area; GCS, global circumferential strain, GLS, global longitudinal strain; mPAP, mean pulmonary artery pressure; NT-proBNP, N-terminal fragment of pro-brain natriuretic peptide; NYHA, New York Heart Association; PVR, pulmonary vascular resistance; RV, right ventricle; RVMASS, right ventricular mass; SVi, stroke volume indexed to body surface area.

**Table 3 diagnostics-12-03183-t003:** Spearman correlation (rho) of strain and physiological parameters in the whole cohort and before/after PEA.

Association	All Patients ^1^		Before PEA		After PEA	
	rho	*p*	rho	*p*	rho	*p*
GLS vs. Ea ^2^	0.5	0.00001	0.48	0.0004	0.5	0.0002
GCS vs. Ea	0.49	0.00001	0.3	0.03	0.34	0.014
GLS vs. Ees	0.05	0.51	−0.48	0.0005	0.19	0.197
GCS vs. Ees	−0.03	0.66	−0.52	0.0001	−0.1339	0.3537
GLS vs. Ees/Ea	−0.56	0.00001	−0.69	0.00001	−0.47	0.0006
GLS vs. RV EF	−0.53	0.00001	−0.67	0.00001	−0.37	0.008
GCS vs. RV EF	−0.6	0.00001	−0.64	0.00001	−0.51	0.0002
GLS vs. RV EDVi	0.23	0.003	0.39	0.005	−0.038	0.79
GCS vs. RV EDVi	0.27	0.0005	0.27	0.067	0.13	0.351
GLS vs. RV ESVi	0.39	0.00001	0.61	0.00001	0.17	0.24
GCS vs. RV ESVi	0.51	0.00001	0.49	0.0003	0.39	0.005
GLS vs. RV SVi	−0.31	0.00001	−0.48	0.0005	−0.41	0.003
GCS vs. RV SVi	−.23	0.0012	−0.3	0.03	−0.29	0.04
GLS vs. mPAP	0.47	0.00001	0.4	0.0045	0.34	0.016
GCS vs. mPAP	0.48	0.00001	0.18	0.21	0.27	0.05
GLS vs. PVR	0.52	0.00001	0.4	0.0065	0.47	0.001
GCS vs. PVR	0.48	0.00001	0.27	0.07	0.07	0.6
GLS vs. NT-proBNP	0.45	0.00001	0.6	0.00001	0.31	0.04
GCS vs. NT-proBNP	0.37	0.00001	0.34	0.03	0.23	0.12

^1^ Data from 50 patients who had both baseline and follow up; ^2^ Abbreviations: Ea, pulmonary arterial end-systolic pressure/volume ratio; EDVi, end-diastolic volume indexed to body surface area; Ees, right ventricular end-systolic pressure/volume ratio; EF, ejection fraction; ESVi, end-systolic volume indexed to body surface area; GCS, global circumferential strain, GLS, global longitudinal strain; mPAP, mean pulmonary artery pressure; NT-proBNP, N-terminal fragment of pro-brain natriuretic peptide; PVR, pulmonary vascular resistance; RV, right ventricle; SVi, stroke volume indexed to body surface area.

## Data Availability

Data available upon request at the author’s institution.

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
