# Peer review of "Right Ventricular Strain by Magnetic Resonance Feature Tracking Is Largely Afterload-Dependent and Does Not Reflect Contractility: Validation by Combined Volumetry and Invasive Pressure Tracings"

_diagnostics, 2022, doi:10.3390/diagnostics12123183_

Round 1
Reviewer 1 Report
-The paper should be interesting ;;;
-it is a good idea to add a block diagram of the proposed research (step by step);;;
-it is a good idea to add more photos of measurements, sensors + arrows/labels what is what (if any);;;
-What is the result of the analysis?;;
-figures should have high quality. ;;;;;
-text should be formatted;;;;
-please add photos of the application of the proposed research, 2-3 photos ;;;
-what will society have from the paper?;;
-please compare the advantages/disadvantages of other approaches etc.;;;
-references should be from the web of science 2020-2022 (50% of all references, 30 references at least);;;
-Conclusion: point out what have you done;;;;
-please add some sentences about future work;;;
Author Response
please see attachement

Reviewer 2 Report
This article features the valuable clinical findings! Sound methodology and great graphs make the material easy to understand
Limitations of the study are clearly labeled. Great article overall
Author Response
please see attachement
Round 2
Reviewer 1 Report
Fig. 1 should be better quality;;;
Figure 2 please add arrows + labels what is what;;;
Figure 4 and 5 - labels, fonts should be bigger
Author Response
We thank the reviewer for attentively reading the revised form of our paper. We have made further improvements of our figures and hope they are now satisfactory for the reviewer in their present form.
We have enlarged fig. 1 and increased it to 300dpi
We have added arrows to fig. 2 and have added an arrow legend in the accompanying text
We have enlarged fig. 3 to 5, the size of the fonts itself is restricted by the statistics application.
Best regards Andreas Rolf